# Four Decades of Laccase Research for Wastewater Treatment: Insights from Bibliometric Analysis

**DOI:** 10.3390/ijerph20010308

**Published:** 2022-12-25

**Authors:** Kana Puspita, Williams Chiari, Syahrun N. Abdulmadjid, Rinaldi Idroes, Muhammad Iqhrammullah

**Affiliations:** 1Department of Chemistry Education, Faculty of Education and Teacher Training, Universitas Syiah Kuala, Banda Aceh 23111, Indonesia; 2Department of Mathematics, Faculty of Mathematics and Natural Sciences, Universitas Syiah Kuala, Banda Aceh 23111, Indonesia; 3Innovative Sustainability Lab, PT. Biham Riset dan Edukasi, Banda Aceh 23243, Indonesia; 4Department of Physics, Faculty of Mathematics and Natural Sciences, Universitas Syiah Kuala, Banda Aceh 23111, Indonesia; 5Department of Chemistry, Faculty of Mathematics and Natural Sciences, Universitas Syiah Kuala, Banda Aceh 23111, Indonesia; 6Department of Pharmacy, Faculty of Mathematics and Natural Sciences, Universitas Syiah Kuala, Banda Aceh 23111, Indonesia

**Keywords:** organic pollutant, emerging contaminant, Scopus, network visualization

## Abstract

Increasing trends of environmental pollution and emerging contaminants from anthropogenic activities have urged researchers to develop innovative strategies in wastewater management, including those using the biocatalyst laccase (EC 1.10.3.2). Laccase works effectively against a variety of substrates ranging from phenolic to non-phenolic compounds which only require molecular oxygen to be later reduced to H_2_O as the final product. In this study, we performed a bibliometric analysis on the metadata of literature acquired through the Scopus database (24 October 2022) with keyword combination “Laccase” AND “Pollutant” OR “Wastewater”. The included publications were filtered based on year of publication (1978–2022), types of articles (original research articles and review articles) and language (English). The metadata was then exported in a CSV (.csv) file and visualized on VosViewer software. A total of 1865 publications were identified, 90.9% of which were original research articles and the remaining 9.1% were review articles. Most of the authors were from China (*n* = 416; 22.3%) and India (*n* = 276; 14.79%). In the case of subject area, ‘Environmental Science’ emerged with the highest published documents (*n* = 1053; 56.46%). The identified papers mostly cover laccase activity in degrading pollutants, and chitosan, which can be exploited for the immobilization. We encourage more research on laccase-assisted wastewater treatment, especially in terms of collaborations among organizations.

## 1. Introduction

The challenge to achieve a sustainable planet has become more difficult with the increase of population resulting in increased hazardous contaminant in water environments [1,2]. Moreover, municipal waste and pharmaceutical waste, especially those deriving from the coronavirus disease 2019 (COVID-19) pandemic, have alarmed researchers in environmental management-related fields due to their harmfulness and persistence [3,4]. This environmental threat has affected the innovation on wastewater treatments, including those using renewable and nontoxic oxidative degrading enzyme—laccase [5,6].

The mechanism of laccase in degrading organic pollutants is dependent on redox reactions involving molecular oxygen and the reduction of Cu^2+^ to Cu^+^ nucleus of this enzyme. Laccase was first introduced as an unspecific enzyme observed in *Rhus vernicifera* sap in Japan [7], and its name was only given a decade later following the success of its isolation and purification [8]. Researchers have isolated the enzyme from various plant, fungal, and bacterial species [9,10,11]. Strategies involved in laccase-related research for wastewater treatment include the isolation of laccase from new sources. *Trametes versicolor* is a popular source of enzyme with high catalytic degrading activity against dying agent contaminants [12]. Other than the fungus, *Pleurotus ostreatus* [13,14] and marine fungi (such as *Trichoderma asperellum, Stemphylium lucomagnoense,* and *Aspergillus nidulans* [15]) have been reported to produce laccase with the enzymatic degradation activity against organic pollutants. The exploration of laccase-producing microbes has been carried out on wastewater as well [16,17,18].

Rapid progress in genetic modifications have allowed researchers to employ these as strategies to obtain more efficacious laccase. The use of metagenomic data collected from tannery wastewater to obtain thermo-halotolerant laccase has been reported [19]. UV light-induced mutation has also been used to isolate higher efficient laccase from *P. ostreatus* mutant [20]. Mutation involving polymerase chain reaction has been involved in a research protocol to produce laccase from *Bacillus licheniformis* mutant [21]. Furthermore, innovation in laccase-themed research also includes the enzyme immobilization to promote its stability and reusability, which is significant to reduce the wastewater treatment cost [22,23,24]. 

Our work aimed to provide the research landscape of laccase-assisted wastewater treatment based on bibliometric analysis. By utilizing the metadata of published studies obtained from a scientific database (i.e., Scopus), a bibliometric analysis could be performed to map the research trends, especially those from newly emerging fields. The occurrence and incidence of environmental pollution have been analysed using bibliometric analysis against different contexts (i.e., mining effluent) [25,26]. In our previous project, we have employed the same analysis to reveal the trend of COVID-19-related polymer research in [27]. Herein, we present a bibliometric analysis on laccase-assisted wastewater treatment which has not been reported previously.

## 2. Methods

### 2.1. Study Design and Searching Strategy

This study employed the Scopus database for literature search reporting laccase utilization in wastewater treatment. The keywords used in the Scopus database search were as follows: (“Laccase” AND (“Pollutant” OR “Wastewater”)) without constrains set in the publication year. The search resulted in 2073 papers, and was then filtered to include only finalized publications (including original articles, review articles, and any types of publication). Only papers written in English were included. Thereafter, we obtained 1865 papers included in this study. The workflow diagram of the literature search performed in this study has been presented in Figure 1. All metadata were downloaded from Scopus database as a CSV (.csv) file. The metadata were visualized using VosViewer 1.6.17. Crosschecking was conducted to avoid potential disambiguation in the exported data.

### 2.2. Data Analysis and Visualization

Microsoft Excel 2016 was used to analyze the selected papers and to export the graphs and tables needed to present the paper type, source journals, top 10 organizations or countries and the most cited papers. Data were double-checked using Biblioshiny (https://www.bibliometrix.org/, accessed on 17 December 2022). Furthermore, VosViewer 1.6.17. was used to conduct the bibliometric analysis such as the co-authorship/co-occurrence analysis and to generate data visualization.

## 3. Results

### 3.1. Characteristic of Identified Papers on Laccase and Pollutant or Wastewater

A total of 1865 papers of original articles (*n* = 1697, 90.9%) and review articles (*n* = 168, 9.1%) regarding laccase and pollutant/wastewater were identified through a Scopus database search with the publication year ranging from 1978 to 2022. A majority of the identified papers (*n* = 1413, 75.76%) were published in the past decade (2012–2022), showing the increase of the study of lacasse with regard to its use in dealing with pollutants or wastewater (Figure 2).

Environmental Science (*n* = 1052, 56.4%), Biochemistry, Genetics and Molecular Biology (*n* = 581, 31.15%), and Chemical Engineering (*n* = 561, 30.08%) were the most studied subject areas among papers related to laccase and pollutants or wastewater (Table 1). A total of 165 journals published 1865 papers related to laccase and pollutants or wastewater, with only 16 journals publishing more than 20 papers (Table 2). These 16 journals (9.69% of total journals) published 701 papers (37.58% of total papers). Bioresource Technology (*n* = 108, 5.79%) was the leading journal publishing the most papers, followed by the Journal of Hazardous Materials (*n* = 95, 5.09%) and Chemosphere (*n* = 89, 4.77%). These three journals were also the only ones to publish more than 50 papers, contributing to 15.65% of total papers published (Table 2).

### 3.2. Top 10 Organizations, Countries, and Funding Sources

From the analysis, Shivaji University (*n* = 147, 7.88%), Jiangnan University (*n* = 95, 5.09%), and Tianjin University (*n* = 92, 4.93%) were revealed as the three most productive organizations, as well as the only organizations with more than 90 published papers (Table 3). Almost a half of the papers were published in China (*n* = 416,22.3%), India (*n* = 276, 14.79%) and Spain (*n* = 162, 8.68%) combined (Table 4). Finally, the top three funding sources were the National Natural Science Foundation of China (*n* = 225, 13.67%), the Natural Sciences and Engineering Research Council of Canada (*n* = 48, 2.57%), and the European Regional Development Fund (*n* = 41, 2.19%) (Table 5).

### 3.3. Mostly Cited Article

There was a total of 74,833 citations for 1865 papers, averaging 40.12 citations per paper and 1700.75 citations per year (1978–2022). The most cited paper was *Biodegradation aspects of Polycyclic Aromatic Hydrocarbons (PAHs): A review,* authored by Haritash, A.K. and Kaushik, C.P. in 2009, which was cited in 2072 papers, almost twice as many as *Technologies for the removal of phenol from fluid streams: A short review of recent developments,* written by Busca, G., et al. in 2008, cited in 1046 papers. There were only three papers cited more than 1000 times. The trend in citations continued to rise across the years, and peaked in 2022 with 11249 citations. The data of the top 10 most cited papers have been presented (Table 6).

### 3.4. Results from Co-Authorship Networking Analysis

A co-authorship analysis was conducted to evaluate the relationship among authors, organizations and countries according to the number of papers they published together. This analysis indicates the trends of collaborations and identifies the leading authors, organizations, and countries in terms of the related study, which in this case was laccase and pollutant/wastewater. Using VosViewer software, a visualization map will be generated which will also show nodes of various sizes representing the author, organization, or country. These nodes will also be connected by a line called Link Strength (LS), where thickness determines the link strength corresponding to the number of co-authored papers and how strong the collaboration is between the two authors, organizations, or countries. All links of a node will be accumulated as Total Link Strength (TLS), representing how strong the author, organization, or country is connected to others [38].

To analyze the authors’ co-authorship, the minimum number of papers per author was set to three and the minimum times of citations of an author was set to 0. Of 5573 identified authors, 689 met the thresholds. The network visualization and overlay visualization maps are shown in Figure 3. Zhang, X. was the leading author in terms of Total Link Strength (Documents = 35, citations = 827, TLS = 125), followed by Govindwar, S. P. (Documents = 36, citations = 2453, TLS = 98), Caminal, G. (Documents = 25, citations = 1473, TLS = 97) and Nghiem, L.D. (Documents = 18, citations = 915, TLS = 97). Iqbal, H.M.N. and Bilal X. (LS = 18) had the most collaborations in publishing the papers.

Meanwhile, an organizational’ co-authorship analysis was conducted on 4165 identified organizations, with the minimum number of papers per organization set to three and the minimum times of citation of an organization was set to 0. Of 4165 organizations, 97 met the thresholds, however only 11 organizations were found to be linked to each other. The network visualization and overlay visualization maps are shown in Figure 4. The School of Life Science and Food Engineering, Huaiyin Institute of Technology, Huaian (Documents = 18, citations = 1045, TLS = 25) and School of Civil, Environmental and Chemical Engineering, RMIT University Melbourne, Australia (Documents = 6, citations = 472, TLS = 15) were the leading organizations. The lack of connected nodes to be shown on the visualization map shows that there is still a lack of collaboration among organizations, with o the School of Life Science and Food Engineering, Huaiyin Institute of Technology, Huaian (China) and Technologico de Monterrey (Mexico) collaborating the most (LS = 5).

Finally, countries’ co-authorship analysis was conducted on 89 identified countries, with the minimum number of papers per country set to three and the minimum times of citations of a country set to 0. The results of this analysis have been presented in Figure 5. Of 89 countries, 65 met the thresholds. The top three countries were China (Documents = 416, citations = 11,579, TLS = 167), India (Documents = 276, citations = 13,240, TLS = 127) and Spain (Documents = 162, citations = 10,551, TLS = 138). There were only four countries that published more than 100 papers, including the aforementioned top three and the United States. India and South Korea had the most collaborations (LS = 31), followed by China and the United States (LS = 25) and China and Mexico (LS = 20).

### 3.5. Keyword Co-Occurrence Analysis

Keyword co-occurrence analysis is a visualization approach to indicate how frequent and high the connected terms or words used in papers are related to a specific field of research. This analysis is also able to identify the development trend of the research focus, generally grouping the keywords into clusters. Using VosViewer software, each node indicates the keyword, while the link line connecting the nodes indicate the keyword relationship and their use on a paper. Both nodes and link lines are varied in size, showing the popularity of the keyword.

For this study, a total of 3777 keywords were identified, with the minimum number of occurrences of a keyword set to 10. Of 3777 keywords, 100 met the threshold. The most used main keyword in the study on laccase and pollutant/wastewater was laccase (occurrences = 753), while the other included keywords were biodegradation (occurrences = 189), decolorization (occurrences = 165), bioremediation (occurrences = 141) and immobilization (occurrences = 102). These five keywords were also the only five keywords to pass the 100 occurrences-mark. The network visualization shown in Figure 6a shows different colours based on the keywords’ closeness to a certain research theme. The keywords coloured in red indicate their closeness to laccase immobilization; in blue—biodegradation; in green—redox enzymes similar to laccase (i.e., peroxidase); in purple—ligninolytic enzyme; in yellow—laccase effectiveness against various pollutants based on decolorization; and in cyan—types of pollutant which could be effectively removed by laccase (including lignin, catechol, phenolic compounds, and olive oil mill wastewater). Nonetheless, it is not clear what research theme is indicated by the keywords cluster in orange colour.

The density visualization was altered to show the keyword change trends across the past 20 years, as the default range (1978–2022) did not show any significant change over the keywords use (Figure 6b). The analysis revealed that in the last 10 years, laccase-assisted wastewater treatment mostly involved the enzyme immobilization indicated by its performance in biodegradation and decolorization. Chitosan, a supporting polymer material, was found as the most connected keyword (Figure 6b).

Density visualization of the keywords that occurred in laccase-assisted wastewater treatment research has been presented in Figure 6c. The analysis indicated that the keywords in red occurred in papers with the highest density, followed by keywords grouped in yellow, green, cyan, and blue. The analysis suggested that biodegradation was the keyword with the highest density. Immobilization, decolorization, and biosorption were the only keywords presented in yellow coloured zone. Types of pollutant are found localized in the green coloured area, namely bisphenol, dye, wastewater, and azo dye.

## 4. Discussion

Laccases (EC 1.10.3.2), a member of the multicopper oxidases family, have been reported for their efficacy in removing phenolic or non-phenolic contaminating agents which can be obtained through an isolation from plants, bacterial, or fungal species [39]. Despite its variety in structure (as isolated from different organisms), they have highly conserved region amino acids which are covalently bonded with the copper centres [40,41]. Laccase induces oxidative reaction against a phenolic molecule involving one electron from one of the enzyme’s Cu centres (Cu^2+^/Cu^+^) to produce a phenolic radical. This initiates a long chain reaction which eventually induces the cleavage of the organic contaminant and the reduction of molecular O_2_ into H_2_O [42]. In terms of efficiency, an immobilized laccase was reported to yield a 90% or higher decontamination rate [22,43,44,45,46]. As many strategies have been employed to improve the utilization of laccase in wastewater treatment, we have performed a bibliometric analysis on a big and reliable scientific database, Scopus [47].

A bibliometric analysis was performed on the 1865 identified papers reporting the utility of laccase in pollutant or wastewater treatment. This study aimed to provide a thorough assessment of the study status according to the source journals, leading organizations, countries and the collaborations between each other. *Bioresearch Technology* published the highest number of papers about laccase and wastewater or pollutants (5.79% of 1865 papers), followed by the *Journal of Hazardous Materials* (5.09% of the papers). Govindwar, S.P. published the most number of papers and also recorded the highest number of citations. An author co-authorship analysis showed that Iqbal, H.M.N. and Bilal, M. had the highest number of collaborations between each other.

The organizations publishing papers on laccase-assisted wastewater treatment were centred in China and India, with Shivaji University (India), Jiangnan University (China), and Tianjin University (China) producing the most papers compared to other institutions. China published the largest number of papers (22.3% of papers), almost twice as many as India (14.79% of papers). These two countries were also the only ones who managed to pass the 10% mark. China’s dominance may occur due to its large funding sources, one of them being the National Natural Science Foundation of China, which was also the largest funding source with regard to papers about laccase and pollutants or wastewater. The aforementioned funding source contributed to 13.67% of 1865 papers, followed by the Natural Sciences and Engineering Research Council of Canada, which contributed funding to 2.57% of papers. Of the 10 top funding sources, both Asia and Europe had three funding sources, while the rest were two funding sources from both North America and South America. The number of collaborations remained an issue among organizations, with only 0.26% of 4165 organizations having collaborations.

Between 1978 and 2022, publications on laccase and pollutant or wastewater reached a total of 1865, of which 90.9% were research articles and the remaining 9.1% were review articles. The total number of identified papers is still significantly smaller compared to the 14,057 total papers that studied laccase (a search was conducted using the same term excluding the specific keywords used in this bibliometric analysis). This shows the dominance of experimental work on laccase and pollutants or wastewater. However, the top papers found in this present study (indicated by the number of citations) were mostly review papers. Whilst most papers specifically discussed laccase activity in degrading pollutants, a paper authored by Rodríguez Couto and Toca Herrera highlighted chitin and chitosan characterizations [31]. We also found that chitosan as the only biopolymer shown in the keyword networking analysis. Taken together, chitosan is the most studied polymeric support for laccase immobilization. This is due to the fact that chitosan has functional groups that can be manipulated for the immobilization reactions or involved interactions [48,49,50,51]. Moreover, chitosan could play a part in contaminant removal via the adsorption on the polymer surface [52,53,54].

A visual networking analysis of the keywords on papers published from 1978 revealed that biodegradation has the highest density, followed by immobilization, decolorization, and biosorption. This suggests that the research is dominated by the utility of non-isolated laccase for the wastewater treatment [55,56,57]. As mentioned previously, immobilization could enhance the removal efficiency whilst improving its working condition ranges and reusability [55,58,59,60]. Biosorption is a keyword usually used to explain the adsorption phenomenon on bio-based materials. Coconut and rice husks are among the growing substrates that have been reported to immobilize *T. versicolor* and could also perform biosorption [61,62,63].

## 5. Challenges and Future Prospects

To get an insight on the constrains of laccase-assisted wastewater treatment, we performed a literature search on review articles using the same previously used keyword combination, but the search was refined to only include those published between 2018 and 2023. Challenges and recommended topics for future research have been presented in Table 7. The high-costs of enzyme production along with its stability and reusability limit its application in wastewater treatment [64,65,66,67,68,69,70,71]. Research on gene editing and molecular engineering could overcome the stability and improve the production yield of the enzyme [64,65,66,67,68,69,70,71]. Using a low-cost solid substrate would also reduce the production cost [13,72,73,74]. Immobilizing laccase onto a supporting material could enhance the stability, reusability and contaminant removal performance [70,71,75,76]. However, laccase immobilization creates other challenges such as enzyme leakage, catalytic site blockage (hence reducing enzymatic activities), and additional production costs [64,71,75]. Therefore, it is important to consider the appropriate immobilization technique and the type of supporting materials. Another challenge in laccase research for contaminant removal is our lack of understanding regarding its catalytic degradation mechanisms, especially because the mechanism is dependent on the type of the contaminants. Screening the enzymatic activity against different pollutants could be assisted by an in silico approach [6,66,70,77].

## 6. Strengths and Limitations

Being the first study to research the bibliometric analysis on the study of laccase and pollutants or wastewater trends, this study used VosViewer to analyse the inputted data searched from the Scopus database, offering more accountability and objectivity. It is worth mentioning that we did not include book chapters, conference paper/reviews, editorial notes, erratum, letters, short surveys, or retracted documents. This study was also based on the Scopus database, and did not include other databases such as Web of Science (WoS). We also found difficulties in the visual networking analysis, including the incomplete name of affiliation in the visualization, an issue which cannot be resolved by the user manually.

## 7. Conclusions 

Laccase utilization in degrading pollutants is mostly studied by scientists in China (22.3%) and India (14.8%). The leading journal publishing research about laccase utilization in wastewater treatment was *Bioresource Technology*. Environmental Science was the most reported subject area, indicating the urgency of laccase use on wastewater treatment. Laccase immobilization is on the research spotlight, where the high occurrence of the keyword ‘chitosan’ has been found. This research, however, was still lacking in terms of collaboration between organizations, which needed to be strengthened to further ensure the betterment in terms of treating wastewater or pollutants.

There are also several recommendations for the bibliometric analysis. Despite being user friendly, the visualization using VosViewer software restricts the user to editing the name in the presentation. In our case, we found the affiliation being presented with an incomplete name, and there was no option for manual editing. There should be a tool to analyze the quality of the paper rather than being based solely on the quantity. More importantly, the bibliometric analysis should be validated with a different approach, and in our case we used Biblioshiny.

## Figures and Tables

**Figure 1 ijerph-20-00308-f001:**
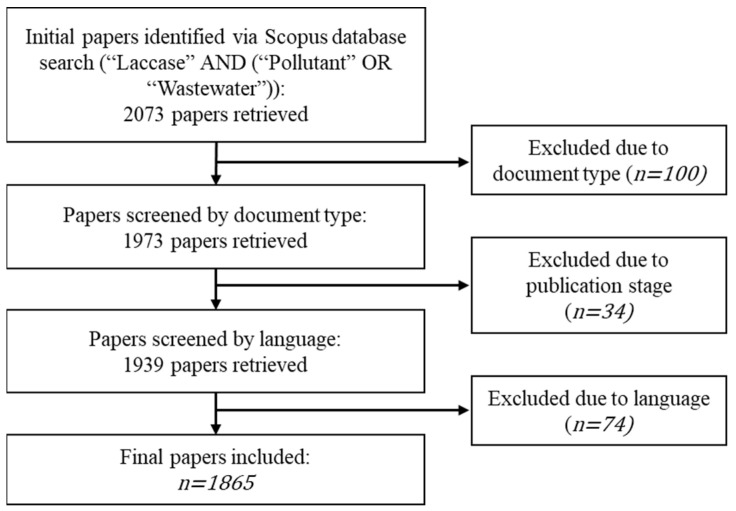
Flowchart of included studies on laccase utilization in wastewater treatment.

**Figure 2 ijerph-20-00308-f002:**
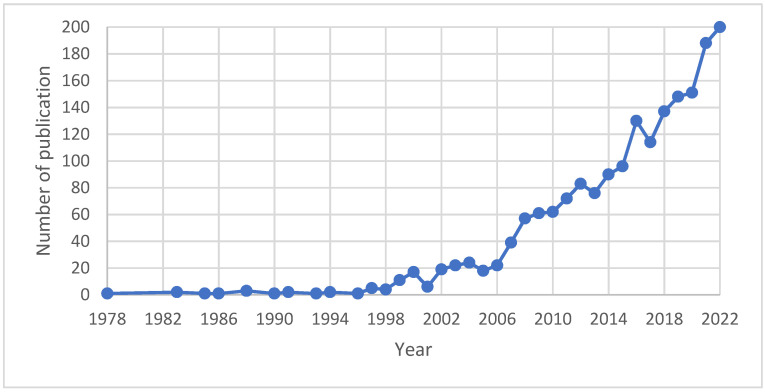
Publication trend of laccase research on pollutant or wastewater.

**Figure 3 ijerph-20-00308-f003:**
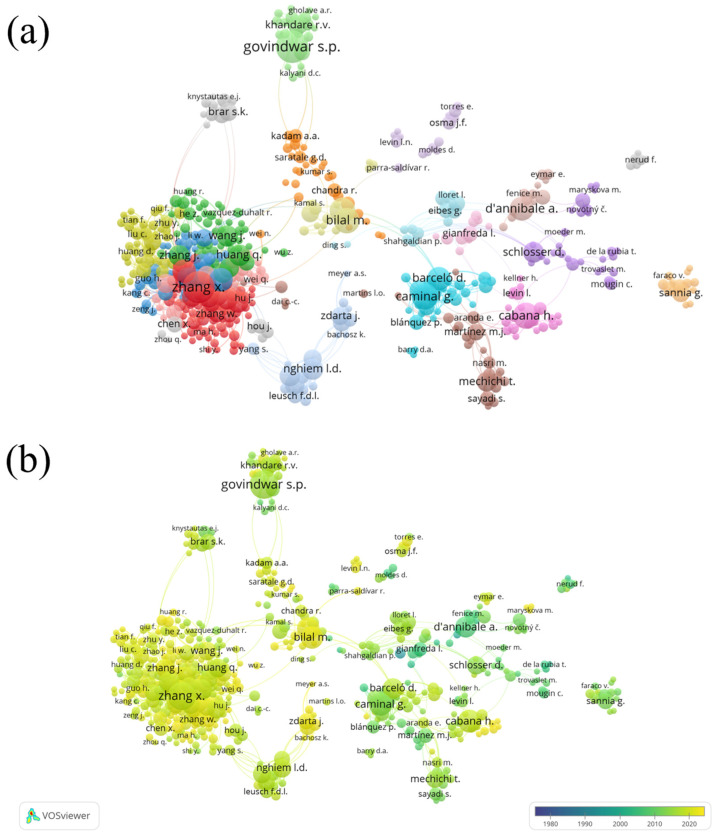
(**a**) Network visualization of authors’ co-authorship (weights: documents). (**b**) Overlay visualization of authors’ co-authorship in 1978–2022 (weights: documents; scores: average publications per year).

**Figure 4 ijerph-20-00308-f004:**
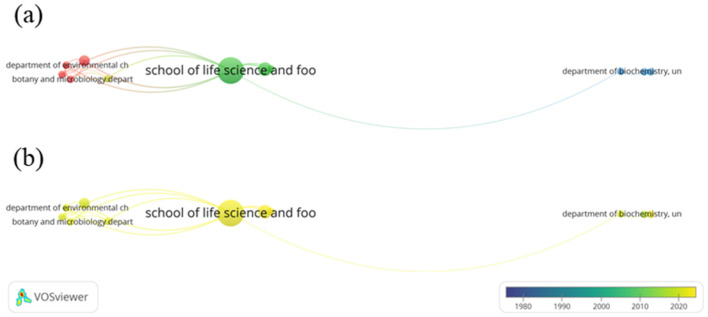
(**a**) Network visualization of organizations’ co-authorship (weights: documents). (**b**) Overlay visualization of organizations’ co-authorship in 1978–2022 (weights: documents; scores: average publications per year). Only connected organizations were shown.

**Figure 5 ijerph-20-00308-f005:**
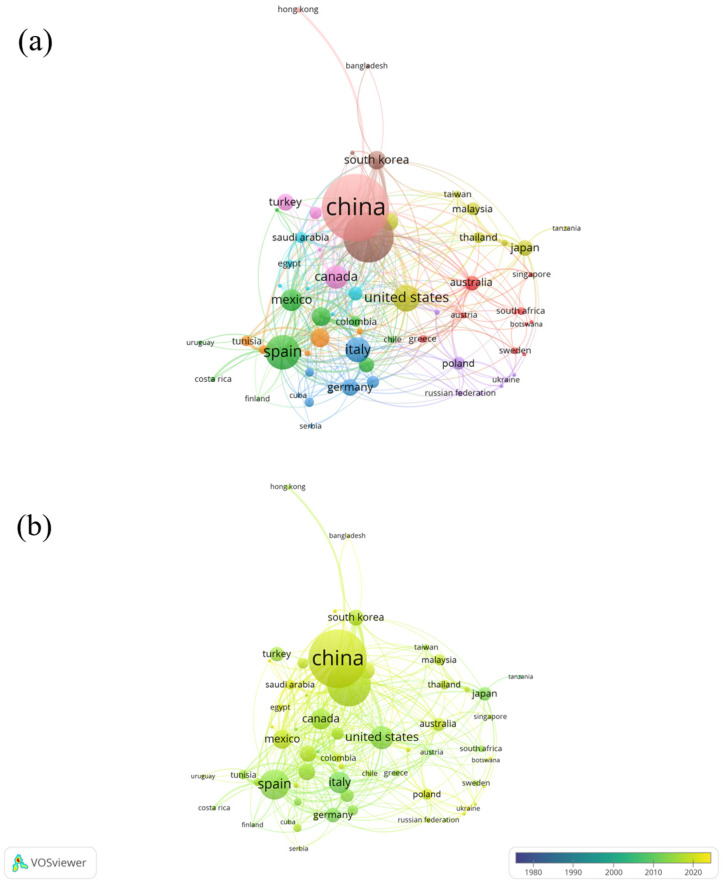
(**a**) Network Visualization of countries’ co-authorship (weights: documents). (**b**) Overlay visualization of countries’ co-authorship in 1978–2022 (weights: documents; scores: average publications per year).

**Figure 6 ijerph-20-00308-f006:**
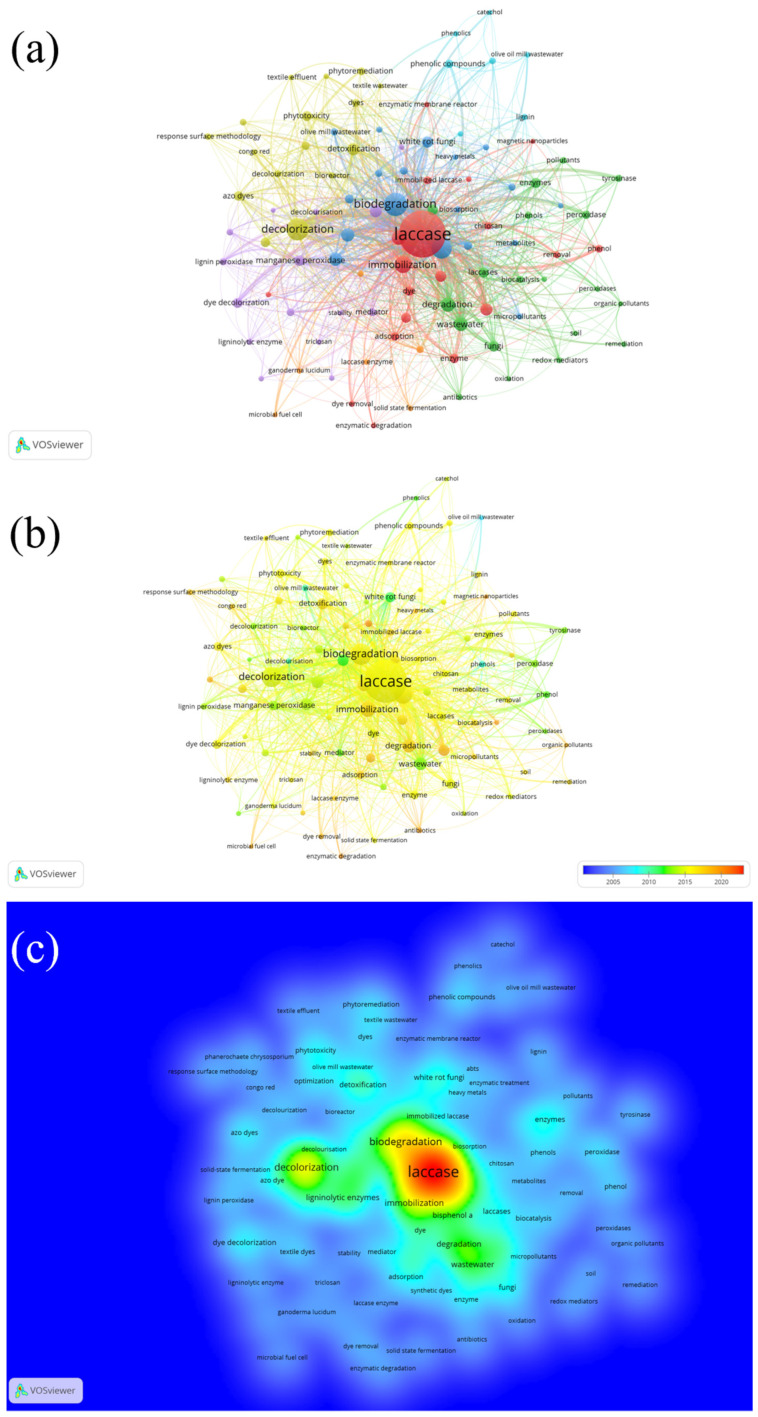
(**a**) Network Visualization of keywords’ co-occurrences (weights: occurrences). (**b**) Overlay visualization of keywords’ co-occurrences in 2002–2022 (weights: occurrences; scores: average publications per year). (**c**) Density visualization of keywords’ co-occurrences (weights: occurrences).

**Table 1 ijerph-20-00308-t001:** Top 10 subject areas related to laccase-assisted wastewater treatment studies.

Rank	Subject Area	Number of Papers
1	Environmental Science	1052
2	Biochemistry, Genetics and Molecular Biology	581
3	Chemical Engineering	561
4	Immunology and Microbiology	445
5	Chemistry	350
6	Energy	188
7	Agricultural and Biological Sciences	172
8	Engineering	160
9	Medicine	144
10	Materials Science	140

**Table 2 ijerph-20-00308-t002:** Top 10 journals publishing articles related to laccase-assisted wastewater treatment.

Rank	Journal	Number of Papers	Number of Citations
1	Bioresource Technology	108	6984
2	Journal of Hazardous Materials	95	7042
3	Chemosphere	89	4157
4	Environmental Science And Pollution Research	49	1153
5	Science of The Total Environment	47	2670
6	International Biodeterioration and Biodegradation	38	2230
7	Applied Microbiology and Biotechnology	35	2201
8	Journal of Environmental Management	35	2076
9	Enzyme and Microbial Technology	30	2801
10	Ecotoxicology and Environmental Safety	27	631

**Table 3 ijerph-20-00308-t003:** Top 10 organizations working on laccase-assisted wastewater treatment.

Rank	Organization	Country	Number of Papers
1	Shivaji University	India	147
2	Jiangnan University	China	95
3	Tianjin University	China	92
4	Hunan University	China	85
5	Universitat Autònoma de Barcelona	Spain	79
6	University of Wollongong	Australia	62
7	Université du Québec	Canada	61
8	Poznan University of Technology	Poland	60
8	Tehran University of Medical Sciences	Iran	60
10	Wuhan University of Science and Technology	China	58

**Table 4 ijerph-20-00308-t004:** Top 10 countries of the authors’ affiliation of published articles reporting on laccase–assisted wastewater treatment.

Rank	Country	Number of Papers	Citations
1	China	416	11,579
2	India	276	13,240
3	Spain	162	10,551
4	United States	112	5039
5	Italy	104	6666
6	Canada	93	5161
7	Mexico	83	3249
8	Brazil	70	3196
9	France	68	2396
10	South Korea	67	2824

**Table 5 ijerph-20-00308-t005:** Top 10 funding sources for laccase–assisted wastewater treatment studies.

Rank	Funding Source	Number of Papers
1	National Natural Science Foundation of China	255
2	Natural Sciences and Engineering Research Council of Canada	48
3	European Regional Development Fund	41
4	Consejo Nacional de Ciencia y Tecnología	39
5	Conselho Nacional de Desenvolvimento Científico e Tecnológico	36
6	University Grants Commission	35
7	European Commission	34
8	Fundamental Research Funds for the Central Universities	30
9	Coordenação de Aperfeiçoamento de Pessoal de Nível Superior	29
10	Ministerio de Economía y Competitividad	29

**Table 6 ijerph-20-00308-t006:** Top 10 most cited papers reporting on laccase-assisted wastewater treatment.

Rank	Title	Author(s)	Year	Citations	Ref.
1	Biodegradation aspects of Polycyclic Aromatic Hydrocarbons (PAHs): A review	A.K. Haritash & C.P. Kaushik	2009	2072	[28]
2	Technologies for the removal of phenol from fluid streams: A short review of recent developments	G. Busca, S. Berardinelli, C. Resini, & L. Arrighi	2008	1046	[29]
3	Industrial and biotechnological applications of laccases: A review	S.R. Couto & J.L.T. Herrera	2006	1030	[30]
4	Functional characterization of chitin and chitosan	S. Rodríguez Couto & J.L. Toca Herrera	2009	894	[31]
5	Microbial decolourisation and degradation of textile dyes	G. McMullan, C. Meehan, A. Conneely, N. Kirby, T. Robinson, P. Nigam, I. Banat, R. Marchant, & W. Smyth	2001	751	[32]
6	Potential applications of oxidative enzymes and phenoloxidase-like compounds in wastewater and soil treatment: A review	N. Durán & E. Esposito	2000	734	[33]
7	Applications of laccases and tyrosinases (phenoloxidases) immobilized on different supports: A review	N. Durán, M. A. Rosa, A. D’Annibale, & L. Gianfreda	2002	667	[34]
8	Decolorization and detoxification of textile dyes with a laccase from *Trametes hirsuta*	E. Abadulla, T. Tzanov, S. Costa, K-H. Robra, A. Cavaco-Paulo, & G. M. Gübitz	2000	638	[35]
9	Decolorization of dye wastewaters by biosorbents: A review	A.V. Srinivasan & T. Viraraghavan	2010	622	[36]
10	Untapped potential: Exploiting fungi in bioremediation of hazardous chemicals	H. Harms, D. Schlosser, & L.Y. Wick	2011	589	[37]

**Table 7 ijerph-20-00308-t007:** Challenges and future research agenda for laccase-assisted wastewater treatment.

Challenge	Proposed Research Topic	Refs.
High-cost enzyme production Low enzyme stability and reusability	Mutant strains DNA recombination Low-cost substrate Increasing the production yield Site-directed mutagenesis Glycosylation modification	[64,65,66,67,68,69,70,71]
Unclear catalytic mechanism due to a large variety of contaminants	Elucidation of the catalytic mechanism In silico methods	[6,66,70,77]
Harsh wastewater treatment operating conditions	Bioreactor design Immobilization	[69,70]
High-cost of immobilization	Low-cost supports	[6,78]
Reduced enzyme activity after immobilization	Selecting and developing immobilization techniques and supporting materials Immobilization with adsorbent and photocatalyst	[64,71,75]
Improving contaminant removal performance	Immobilization with adsorbent and photocatalyst Combination with nanoparticle Employing magnetic materials Gene editing and mutagenesis	[70,71,75,76]
Enzyme leakage	Selecting and developing immobilization techniques and supporting materials	[78]

## Data Availability

Underlying data are made available by request to the corresponding author.

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
