# Peer review of "Four Decades of Laccase Research for Wastewater Treatment: Insights from Bibliometric Analysis"

_ijerph, 2022, doi:10.3390/ijerph20010308_

Round 1

Reviewer 1 Report

In this review, the authors have performed a bibliometric analysis on the metadata of literatures about the studies of laccase. It is not an analysis of the research content, but merely the statics to show which country or person or funding agency published the most papers. I do not find any novelty in this kind of analysis since all data is available through library database.

Regarding the content of this paper, I have some comments listed below.

1) Figure 3-6 is not clear. Too much informations are not presented clearly. These are needed to be redrawn.

2) Regarding table 3, I think it is unuseful to evaluate researcher by the number of their publications. In conclusion, the authors stated that Govindwar, S.P. is the most prolific author because he published more papers than other researchers in this area. I could not find any meaning for a researcher to be prolific. Quality and originility are important than numbers of publications.

Author Response

Dear Reviewer 1

Thank you for taking your time reviewing our manuscript. We have improved our manuscript and hope that it will meet your satisfaction. We thank your concerns pertaining to our works, and we have responded your queries. Please see them below.

  1. Comment: In this review, the authors have performed a bibliometric analysis on the metadata of literatures about the studies of laccase. It is not an analysis of the research content, but merely the statics to show which country or person or funding agency published the most papers. I do not find any novelty in this kind of analysis since all data is available through library database.

Response: Thank you for your concern. This is not a review article, we performed the analyses based on the published article metadata using methodology widely used in bibliometric studies. Therefore, it is not necessarily a review article. Furthermore, despite the long journey of laccase research for wastewater management, no bibliometric studies have been performed. Presentation of authors and countries are useful, so researchers who wish to collaborate know who to approach. Moreover, we have added the challenge and future prospect (Section 5) on this topic.

  1. Comment: Regarding the content of this paper, I have some comments listed below.
  • Figure 3-6 is not clear. Too much informations are not presented clearly. These are needed to be redrawn.

Response: Thank you for your concern. First of all, all images generated from VosViewer is automatically generated, and no manual editing can be done. Therefore, it is not feasible to re-drawn the image by our own. Secondly, we have exported the images in highest definition. Reduction in your side probably due to the conversion process by the system. We will ensure that all images would be presented in high quality when published. Thirdly, we agree with you that small presentation may obscure the data, hence, we have enlarged the image size.

  1. Comment: 2) Regarding table 3, I think it is unuseful to evaluate researcher by the number of their publications. In conclusion, the authors stated that Govindwar, S.P. is the most prolific author because he published more papers than other researchers in this area. I could not find any meaning for a researcher to be prolific. Quality and originility are important than numbers of publications.

Response: As a researcher in scientometric myself, I absolutely agree with this comment. Unfortunately, tools for analyzing “Quality and originility” have not been available in bibliometric methods. However, there is a closer approach to identifying the impact (though not perfect) which is based on H-index. The H-indices were calculated based on Biblioshiny. 

“Data based on the Hirsch index (H-index) are pretty much in line with the publication number. The top three positions was placed by Govindwar, S.P.; Caminal, G.; and Barceló, D. (Table 3).

Please also refer to revised version of Table 3.

We have also included recommendations for the bibliometric studies at the end of the manuscript:

There are also several recommendations for the bibliometric analysis. Despite being user friendly, the visualization using VosViewer software restricts user to edit the name in the presentation. In our case, we found affiliation being presented with incomplete name and there was no option for manual editing. There should be a tool to analyse the quality of the paper rather than based solely on the quantity. More importantly, the bibliometric analysis should be validated with different approach, where in our case, we used biblioshiny.

Reviewer 2 Report

1. The definition of the picture output by the software needs to be improved.

2. The main thrust of our work can be elaborated in more detail.

3. The conclusion should be more detailed and organized and explain each conclusion point by point.

4. The name of the picture should be marked below each picture.

Author Response

Dear Reviewer 2,

Thank you for taking your time reviewing our manuscript. Please see our responses below:

  1. Comment: The definition of the picture output by the software needs to be improved.

Response: Thank you for your concern. We have exported the image with the best definition. I’m afraid that the submission system reduced the image to automatically. We will communicate this issue with the editor during the publication stage.

  1. Comment: The main thrust of our work can be elaborated in more detail.

Response: We have added “5.       Challenge and Future Prospect” to accompanied the bibliometric analysis results.

  1. Comment: The name of the picture should be marked below each picture.

Response: We have edited the pictures and make clear caption below each of them.

Reviewer 3 Report

The manuscript presents a bibliometric analysis on the laccase research for wastewater treatment. As far as the reviewer knows, this may be the first study on this topic, with a simple and clear conclusion; however, it lacks some real insight on the future perspective, research direction, or challenges of laccase research, which can be derived from an in-depth analysis of the literatures, rather than relying on the keywords. 

Some other comments: 

1. Ministry Education of China is not a research organization; it should not be counted here. Please edit.  

2. Please verify whether figure 4 is complete. 

Author Response

Dear Reviewer 3,

Thank you for taking the time reviewing our manuscript. We have worked our manuscript based on your critical comments. Please see below our responses:

  1. Comment: The manuscript presents a bibliometric analysis on the laccase research for wastewater treatment. As far as the reviewer knows, this may be the first study on this topic, with a simple and clear conclusion; however, it lacks some real insight on the future perspective, research direction, or challenges of laccase research, which can be derived from an in-depth analysis of the literatures, rather than relying on the keywords.

Response: Thank you for your insight regarding this issue. We agree that in-depth analysis is required, and for that, we have presented another analysis based on our review of review articles. We confirmed that immobilization is the centre strategy to improve laccase stability and catalytic degradation performance (in line with those suggested by the network visualization analysis). Please see “5. Challenge and Future Prospect” located after discussion.

  1. Comment: Ministry Education of China is not a research organization; it should not be counted here. Please edit.

Response: Thank you for this critical point. We have edited and re-analyzed the data using biblioshiny. Apparently, the analysis is more accurate in classifying the affiliation. We have double-checked everything to ensure the data accuracy. Please see Table 4.

  1. Comment: Please verify whether figure 4 is complete.

Response: We noted the truncated name there, probably because of the bibliometric coding. This issue cannot be resolved manually. We have put this as the limitation, in the hope that the VoSviewer tool would be improved in the future.

Round 2

Reviewer 1 Report

I have reviewed authors' response and revised manuscript.

I think table 3 is not suitable to be included in the manuscript, even with H-index. Total citation and H-index is not a good way to evaluate the contribution of a researcher.

Simple reason is that table 7 is top10 mosted cited papers, which means real impact to the scientific community, but it seems that very few researcher in table 3 appeared in the author list of these 10 influential paper.

So I suggest table 3 to be deleted before publication.

Author Response

"So I suggest table 3 to be deleted before publication."

Response: It make sense regarding your thought about the top authors' contribution by comparing both table. We then decided to remove the table and revise our content accordingly. Thank you for this valuable suggestion. 
